# Changes in the Physicochemical Properties of Chia (*Salvia hispanica* L.) Seeds during Solid-State and Submerged Fermentation and Their Influence on Wheat Bread Quality and Sensory Profile

**DOI:** 10.3390/foods12112093

**Published:** 2023-05-23

**Authors:** Elena Bartkiene, Arnoldas Rimsa, Egle Zokaityte, Vytaute Starkute, Ernestas Mockus, Darius Cernauskas, João Miguel Rocha, Dovile Klupsaite

**Affiliations:** 1Department of Food Safety and Quality, Veterinary Academy, Lithuanian University of Health Sciences, Tilzes Str. 18, LT-47181 Kaunas, Lithuania; elena.bartkiene@lsmuni.lt (E.B.); arnoldas.rimsa@stud.lsmu.lt (A.R.); vytaute.starkute@lsmuni.lt (V.S.); 2Institute of Animal Rearing Technologies, Faculty of Animal Sciences, Lithuanian University of Health Sciences, Tilzes Str. 18, LT-47181 Kaunas, Lithuania; egle.zokaityte@lsmuni.lt (E.Z.); ernestas.mockus@lsmuni.lt (E.M.); 3Food Institute, Kaunas University of Technology, Radvilenu Road 19, LT-50254 Kaunas, Lithuania; darius.cernauskas@ktu.lt; 4CBQF—Centro de Biotecnologia e Química Fina—Laboratório Associado, Escola Superior de Biotecnologia, Universidade Católica Portuguesa, Rua Diogo Botelho 1327, 4169-005 Porto, Portugal; 5LEPABE—Laboratory for Process Engineering, Environment, Biotechnology and Energy, Faculty of Engineering, University of Porto, Rua Dr. Roberto Frias, s/n, 4200-465 Porto, Portugal; 6ALiCE—Associate Laboratory in Chemical Engineering, Faculty of Engineering, University of Porto, Rua Dr. Roberto Frias, s/n, 4200-465 Porto, Portugal

**Keywords:** salba-chia, lacto-fermentation, white bread, acrylamide, biogenic amines, fatty acid, volatile compounds

## Abstract

This study aimed at investigating the impacts of 24 h of either solid-state fermentation (SSF) or submerged fermentation (SMF) with *Lactiplantibacillus plantarum* strain No. 122 on the physico-chemical attributes of chia seeds (CS). Furthermore, this study examined how adding fermented chia seeds (10, 20, and 30% concentrations) affected the properties and sensory profile of wheat bread. Acidity, lactic acid bacteria (LAB) viable counts, biogenic amine (BA), and fatty acid (FA) profiles of fermented chia seeds were analysed. The main quality parameters, acrylamide concentration, FA and volatile compound (VC) profiles, sensory characteristics, and overall acceptability of the obtained breads, were analysed. A decline in the concentration of certain BA and saturated FA and an increase in polyunsaturated FA and omega-3 (ω-3) were found in fermented CS (FCS). The same tendency in the FA profile was observed in both breads, i.e., breads with non-fermented CS (NFCS) or FCS. The quality parameters, VC profile, and sensory attributes of wheat bread were significantly affected by the addition of NFCS or FCS to the main bread formula. All supplemented breads had reduced specific volume and porosity, but SSF chia seeds increased moisture and decreased mass loss after baking. The lowest acrylamide content was found in bread with a 30% concentration of SSF chia seeds (11.5 µg/kg). The overall acceptance of supplemented breads was lower than the control bread, but breads with 10 and 20% SMF chia seed concentrations were still well accepted (on average, 7.4 score). Obtained results highlight that fermentation with *Lp. plantarum* positively contributes to chia seed nutritional value, while incorporation of NFCS and FCS at certain levels results in an improved FA profile, certain sensory attributes, and reduced acrylamide content in wheat bread.

## 1. Introduction

Chia seeds, or *Salviae hispanicae semen*, are classified as pseudocereals and oilseeds that are extensively cultivated in such Central and South American countries as Bolivia, Guatemala, Ecuador, Peru, and Mexico [1]. These seeds have already been approved for use in the food industry in Europe and North America by the European Food Safety Authority (EFSA) and the National Nutrient Database, respectively [2]. Due to their rich chemical composition and health-improving properties, chia seeds are known as a valuable functional ingredient for food production and are considered a “novel food” or a “superfood” [2]. Seeds contain the entire range of essential amino acids, among which glutamic acid is predominant [3]. A high level of essential fatty acids is found in chia seeds, primarily α-linolenic and linolenic acids [4]. Besides their high protein (up to 26%) and dietary fibre (up to 30%) content, chia seeds are also a good source of calcium, phosphorus, magnesium, and zinc, as well as vitamins C, A, and E, and B-group vitamins [5,6]. Approximately 9% of the total chia seed constituents are polyphenolic compounds, which mainly include phenolic acids (rosmarinic, chlorogenic, caffeic, gallic, and ferulic acids) and flavonoids (apigenin, rutoside, quercetin, and kaempferol) [7]. The presence of tannins, carotenoids, sterols, and phytates in chia seeds was also reported [8,9]. Numerous scientific studies reported lipid-lowering, hypoglycaemic, and hepatoprotective functions of *S. hispanicae* seeds, as well as other beneficial effects on inflammatory processes, overweight and obesity, neurodegenerative diseases, and diseases of the gastrointestinal tract [10]. Chia seeds contain a high amount of insoluble and soluble dietary fibres, the intake of which can diminish such health issues as diabetes, coronary heart disease, cancer, and gastrointestinal disorders [5,11].

However, the shells of seeds barely break down during the digestion process, impeding the release and absorption of nutrients [12]. Moreover, as was mentioned before, chia seeds contain such “antinutrients” as oxalates, tannins, and phytates, which can interact with nutrients, preventing their absorption [13].

Fermentation is being used to enhance the nutritional profile and sensory and technological properties of cereals, pseudocereals, and oilseeds, leading to lower anti-nutritional factors as well as higher protein content and antioxidant capacity [14,15]. Lactic acid bacteria (LAB) are commonly employed in the food and feed industries and are generally regarded as safe (GRAS). LAB are well known for their enzymatic, antioxidant, and antimicrobial activities, probiotic properties, and their ability to destroy antinutritional compounds, synthesise vitamins, and improve nutrient absorption [16]. In this regard, fermentation may aid in the decomposition of seed shells and enhance the chia seeds’ digestibility. Additionally, fermentation can affect the chemical composition and antioxidant activity of seeds. In a study by Calvo-Lerma et al. [12], chia and sesame seeds were solid-state fermented (SSF) with the edible fungi *Pleurotus ostreatus*. A decrease in saturated fatty acids and increases in protein, lipid, and polyunsaturated fatty acid contents were observed in fermented chia seeds. Abdel-Aty et al. [17] reported that SSF by the fungus *Trichoderma reesei* significantly increased the total phenolic compound content and antioxidant activity of chia seeds. However, scarce literature is available regarding the influence of SSF or submerged fermentation (SMF) by LAB on chia seed characteristics. There are only a few studies on white or gluten-free bread with chia flour and sourdough produced by *Weissella cibaria* and autochthonous lactobacilli [18,19].

Wheat bread is an everyday food and is deemed significant for the human diet around the world because it is high in such macronutrients as protein and carbohydrates and contains dietary fibres, minerals, vitamins, and antioxidants [20]. However, the nutritional value of wheat bread is highly affected by the type of wheat flour (refined or whole grain) and other ingredients used [21]. Because of the significant quantity of highly digestible carbohydrates in refined wheat bread, greater consumption of this type of product has been associated with a rise in such health issues as diabetes, colon cancer, and chronic cardiovascular diseases [22]. Therefore, lately, there has been a considerable rise in public interest in bread with potential health advantages through the presence of bioactive compounds [23].

Chia seeds and flour were already used as ingredients for bread production [24,25,26]. However, besides improvements in the nutritional profile and antioxidant properties of bread, attention should be paid to the fact that the incorporation of chia seeds and further breadmaking processes may affect the technological and sensory quality of bread, nutrient stability and bioavailability, and the formation of other, unwanted, compounds (e.g., acrylamide, furanic compounds, etc.) [26,27,28,29,30].

The present study aimed to investigate the impact of 24 h of either SSF or SMF with *Lactiplantibacillus plantarum* strain No. 122 on the physico-chemical attributes of chia seeds. Furthermore, this study examined how adding fermented chia seeds (10, 20, and 30% concentrations) affected the properties and sensory profile of wheat bread. The acidity, LAB counts, biogenic amines, and fatty acid profiles of fermented chia seeds were analysed. Produced breads were subjected to assessment of specific volume, shape coefficient, crumb porosity, moisture, mass loss after baking, texture, colour, acrylamide concentration, fatty acid and volatile compound profiles, and sensory properties.

## 2. Materials and Methods

### 2.1. Principal Scheme of the Experiment

The principal scheme of the experiment is given in Figure 1. To evaluate the influence of non-fermented (NFCS) and fermented (FCS) chia seeds on bread quality and sensory profile, different quantities of chia seeds were tested (chiefly 10, 20, and 30% of the flour weight). Wheat bread prepared in the absence of chia seeds was analysed as a bread control.

### 2.2. Chia Seeds and Lactic Acid Bacteria Used for Their Fermentation

Chia (*Salvia hispanica* L.) seeds (composition: protein 21%, fat 31%, total carbohydrates 5% (1% from sugar), and dietary fibre 34%) were obtained from Urtekram Ltd. (Copenhagen, Denmark).

For SMF and SSF of chia seeds, *Lactiplantibacillus plantarum* strain No. 122, acquired from the Lithuanian University of Health Sciences collection (Kaunas, Lithuania), was used. This LAB strain was chosen because *Lp. plantarum* No. 122 was previously isolated from spontaneous rye sourdough and showed good tolerance to low pH conditions and antibacterial and antifungal activities. Characteristics of the used No. 122 strain are reported by Bartkiene et al. [31].

#### 2.2.1. Chia Seeds Fermentation

Before the experiment, *Lp. plantarum* No. 122 was incubated and multiplied in De Man, Rogosa, and Sharpe (MRS) broth culture medium (Biolife, Milano, Italy) at 30 °C for 24 h under anaerobiosis. A total of 3 mL of fresh LAB grown on MRS broth [average cell concentration of 9.0 log_10_ colony-forming units (CFU)/mL] were inoculated in 100 g of chia seeds/water mass [1:5 (*w*/*w*) ratio for SSF, 1:10 (*w*/*w*) ratio for SMF]. Afterwards, the chia seeds were fermented in a thermostat (Memmert GmbH Co. KG, Schwabach, Germany) for 24 h at 30 °C. Non-fermented chia seeds were analysed as a seed control.

Before and after fermentation, the pH, total titratable acidity (TTA), viable LAB count, BA concentration, and fatty acid (FA) profile of chia seed samples were analysed.

#### 2.2.2. Analysis Methods of Non-Fermented, Submerged-Fermented, and Solid-State Fermented Chia Seeds

The pH values of samples were evaluated with a pH meter (Inolab 3, Hanna Instruments, Venet, Italy) by inserting the pH electrode into the chia seed samples (which were mixed with water).

The total titratable acidity (TTA) was determined for a 10 g sample homogenised with 90 mL of distilled water, and was expressed as the volume, in mL, of 0.1 mol/L NaOH required to achieve a pH of 8.2 (TTA was assessed in Neiman degrees, °N).

The LAB viable counts were determined according to the method described by Bartkiene et al. [32], and are described in detail in Appendix A.

Sample preparation and determination of the BA, including tryptamine, phenylethylamine, putrescine, cadaverine, histamine, tyramine, spermidine, and spermine, were conducted following the procedure reported by Ben-Gigirey et al. [33] with some modifications, and are described in Appendix A.

The extraction of lipids for FA analysis was done with chloroform/methanol (2:1 *v*/*v*), and FA methyl esters (FAME) were prepared according to the procedure of Pérez-Palacios et al. [34]. All procedures are described in detail in Appendix A.

### 2.3. Wheat Bread Preparation and Analysis Methods of Bread Samples

#### 2.3.1. Wheat Bread Formulation and Preparation Technology

Bread was prepared as it was described by Bartkiene et al. [35]. The bread (white bread) formula consisted of 1.0 kg of refined wheat flour (type 550 D, falling number 350 s, wet gluten 27%, ash 0.68%) obtained from Kauno Grudai Ltd. mill (Kaunas, Lithuania), 1.5% salt (regular, refined table salt, “O‘Sole”, Szczecin, Poland), 3% instant yeast, and 1000 mL of drinking water (room temperature, 22 °C). Control bread samples were prepared without the addition of non-fermented or fermented (SMF or SSF) chia seeds. The tested bread groups were prepared by adding 10, 20, or 30% non-fermented, SMF, or SSF chia seeds to the main recipe. In total, 10 groups of dough and bread were prepared and tested (B_C_—control bread; B_NF10_, B_NF20_, B_NF30_—bread samples with 10, 20, and 30% non-fermented chia seeds, respectively; B_SMF10_, B_SMF20_, B_SMF30_—bread samples with 10, 20, and 30% SMF chia seeds, respectively; and B_SSF10_, B_SSF20_, B_SSF30_—bread samples with 10, 20, and 30% SSF chia seeds, respectively). The dough was mixed for 3 min at a low speed, then for 7 min at a high-speed in a dough mixer (KitchenAid Artisan, OH, USA). Then, the dough was left at 24 ± 2 °C for 15 min of relaxation. Afterwards, the dough was shaped into 375 g loaves, then formed and proofed at 32 ± 2 °C and 80% relative humidity for 60 min. The bread was baked in a deck oven (EKA, Borgoricco PD, Milano, Italy) at 220 °C for 25 min. Three independent batches were baked. After 12 h of cooling at 22 ± 2 °C, bread samples were subjected to analysis of specific volume, shape coefficient, crumb porosity, moisture content, mass loss after baking, texture, acrylamide concentration, crust and crumb colour coordinates, fatty acid and volatile compound profiles, sensory characteristics, and overall acceptability.

#### 2.3.2. Bread Analysis Methods

Bread analysis was performed as it was described by Bartkiene et al. [35]. Bread volume was established using the American Association of Cereal Chemists (AACC) method [36], and the specific volume was calculated as the ratio of volume (cm^3^) to weight (g).

The bread shape coefficient was calculated as the ratio of bread slice width (in mm) to height (in mm).

Bread crumb porosity was evaluated using LST method 1442:(1996) [37].

The moisture content was determined according to the International Association for Cereal Science and Technology (ICC) Standard Method 110/1 [38].

Mass loss after baking was calculated as a percentage by measuring the loaf dough mass before and after baking.

Bread hardness was determined as the energy required for sample deformation using Texture Analyser TA.XT2 (StableMicro Systems Ltd., Godalming, UK). A more detailed description is given in Appendix A.

The acrylamide concentration was determined according to the method of Zhang et al. [39] with some modifications. All procedures are described in detail in Appendix A.

The volatile compounds (VC) were analysed using gas chromatography-mass spectrometry (GC-MS). All procedures are described in detail in Appendix A.

Crust and crumb colour parameters were evaluated using a CIE L*a*b* system (CromaMeter CR-400, Konica Minolta, Tokyo, Japan) [40].

Quantitative descriptive sensory analysis was used for the sensory profile of breads. The intensity of sensory properties was assessed using a 10-point scale, where 0 and 10 indicate the lowest and highest intensity, respectively. Overall acceptability was evaluated using a 10-point Likert scale ranging from 10 (extremely like) to 0 (extremely dislike). The evaluation was carried out according to ISO 11136:2014 [41] and ISO 8586:2012 [42] by 20 females and 10 males aged between 20 and 36 years. All procedures are given in detail in Appendix A.

### 2.4. Statistical Analysis

All results were expressed as the mean values (for bread sensory analysis and acceptability, *n* = 30; for the rest of the parameters, *n* = 3) ± standard error (SE). In order to evaluate the effects of fermentation and different quantities of chia seeds on bread quality parameters, data were analysed by multivariate ANOVA and Tukey HSD tests as post-hoc tests, using R Statistical Software (v4.1.2; R Core Team 2021). Additionally, Pearson correlations were calculated between various parameters. The results were recognised as statistically significant at a *p* level equal to or lower than 0.05 (*p* ≤ 0.05).

## 3. Results and Discussion

### 3.1. Parameters of Non-Fermented and Fermented Chia Seeds

Acidity parameters and LAB viable counts of chia seeds are presented in Table 1. After 24 h of fermentation, the pH of chia seed samples decreased, on average, by 41.8 and 37.1% for SMF and SSF chia samples, respectively. However, in contrast, higher TTA (on average, 18.1% higher) was shown by SSF chia samples in comparison with SMF samples. Significant differences between LAB count in SMF and SSF samples were not established (on average, LAB count in fermented chia samples was 8.78 log_10_ CFU/g). A very strong positive correlation between the TTA and LAB count of samples was found (r = 0.946, *p* = 0.004).

Changes in acidity during the fermentation of chia seeds are elicited by LAB, which produce organic acids, such as lactic and acetic acid, through carbohydrate metabolism [43]. The lower pH values and increased LAB viable counts in SMF samples can be explained by the reduced viscosity of the fermentation medium due to a lower solid to liquid ratio when compared to SSF [44]. In the studies of Bustos et al. [19] and Maidana et al. [18], slightly higher values of pH (4.3 and 5.4) and similar LAB viable counts (9.2 log CFU/g and 7.98 log CFU/mL) for chia flour sourdough (fermented for 24 h) prepared with *Lp. plantarum* C8 and *Lp. plantarum* FUA3165, respectively, were reported. Variations in these parameters may occur due to different fermentation processes as well as LAB enzymatic activity and metabolism of different carbohydrates and plant-derived polysaccharides [31].

Biogenic amine (BA) concentrations (mg/kg) in chia seed samples are shown in Table 2. Putrescine, cadaverine, histamine, and tyramine were not found in all analysed chia samples. Likewise, phenylethylamine and spermine were not established in both (SMF and SSF) fermented samples. The main BA in chia seeds was tryptamine, and its content in samples was, on average, 105.4 mg/kg; additionally, fermentation was not a significant factor for tryptamine content in samples. Spermidine showed a very strong positive correlation with tryptamine (r = 0.986, *p* ≤ 0.001), while spermine showed a strong positive correlation with phenylethylamine (r = 0.805, *p* = 0.009). Additionally, the pH values of samples after 24 h of fermentation showed strong and very strong positive correlations with phenylethylamine and spermine, respectively (r = 0.795, *p* = 0.010 and r = 0.992, *p* ≤ 0.001, respectively). 

Amino acid decarboxylation and the amination and transamination of both aldehydes and ketones lead to the synthesis of biogenic amines; decarboxylase-positive microorganisms are mainly responsible for that phenomenon [45]. BA, in low concentrations, benefit individuals with their metabolic activities but, at greater concentrations, they induce serious food poisoning and other health issues such as allergies, increased blood pressure and proliferation of cells, and brain haemorrhages [46]. The most hazardous BA in foodstuffs are histamine and tyramine [40]. Some countries have regulations concerning the maximum levels of histamine in fish products, but these levels are not unified [46]. In this study, histamine and tyramine were not found in unfermented or fermented chia seeds. Toxic levels of other BA in fermented food have not been fully determined [47]. Such polyamines as spermine and spermidine are naturally found in plant-derived food, while the presence of tryptamine and phenylethylamine in food is usually the result of microbial decarboxylation [48,49]. The accumulation of BA is highly influenced by pH, water-activity, temperature, the presence of oxygen, and redox potential [50]. LAB are the primary microorganisms responsible for the formation of BA in fermented foods [51]. Therefore, it is important to select starter cultures that are incapable of producing these compounds. According to our results, individual BA concentrations after chia seed fermentation did not increase. Moreover, the contents of phenylethylamine, spermidine, and spermine were reduced. This indicates the proper selection of the LAB strain and its possible BA-degrading activity. Indeed, the ability of *Lp. plantarum* to degrade BA was already reported [52].

The fatty acid composition (% of the total fat content) of the chia seed samples is given in Table 3. The main FA in chia seeds (non-fermented and fermented seeds) was α-linolenic acid (C18:3 α), and fermented samples showed, on average, 6.50% higher C18:3 α content in comparison with non-fermented samples. Another dominant FA in chia seeds was linoleic acid (C18:2). The latter FA content was reduced after fermentation (in SMF samples, on average, by 5.83%; in SSF samples, on average, by 3.88%). Palmitic acid (C16:0) content in non-fermented and fermented samples was, on average, 6.62%, and fermentation was not a significant factor influencing C16:0 content in chia seeds. The content of stearic acid (C18:0) and octadecenoic acid (C18:1) in both fermented samples was established, and found to be, on average, 27.7 and 27.4% lower, respectively, in comparison with non-fermented samples. *Cis*-11-eicosenoic acid (C20:1) was found only in non-fermented chia samples (0.13% from the total fat content), and the lowest eicosanoic acid (C20:0) content was observed in SMF chia seeds (on average, 24.4% lower in comparison with non-fermented and SSF samples).

Lower saturated fatty acid (SFA) content (on average, 12.7% lower) was found in both fermented samples when compared with non-fermented samples. However, non-fermented samples showed higher ω-6 and ω-9 monounsaturated fatty acid (MUFA) contents in comparison with fermented samples. In contrast to these findings, a higher content of polyunsaturated fatty acids (PUFA) and ω-3 was found in both fermented sample groups (on average, 4.00 and 6.95% higher, respectively).

Very strong positive correlations between the pH of the samples and C18:1, C20:1, MUFA, and ω-9 FA content were established (r = 0.994, *p* ≤ 0.001; r = 0.992, *p* ≤ 0.001; r = 0.987, *p* ≤ 0.001; r = 0.993, *p* ≤ 0.001, respectively). Additionally, the pH of the samples showed moderately positive correlations with C18:0 and C20 content (r = 0.705, *p* ≤ 0.001 and r = 0.790, *p* = 0.011, respectively).

Currently, chia seeds are regarded as one of the top plant sources of ω-3 and α-linolenic FA [53]. The obtained FA profile of chia seeds is in line with those reported in earlier studies [5,54,55]. Similar tendencies regarding SFA, MUFA, and PUFA in fermented chia seeds were also found by Calvo-Lerma et al. [12], who used edible fungi for SSF of chia. They also stated that unsaturated FA (UFA) are on the rise, whereas those with a saturated carbon chain are on the decline in most plant materials fermented with various microorganisms. However, there is no data on the FA profile of chia seeds fermented with LAB strains. The changes in FA profile after chia seed fermentation with *Lp. plantarum* could be related to the presence of such LAB hydrolytic enzymes as esterases and lipases and the ability of LAB to metabolise fatty acids [56,57,58].

### 3.2. Quality Parameters of Produced Bread

#### 3.2.1. Technological Parameters

Wheat bread crumb images are shown in Figure 2. The results of specific volume, shape coefficient, porosity, moisture content, and mass loss after baking analyses are depicted in Table 4. In all cases, bread supplementation with chia seeds reduced bread specific volume, and the lowest specific volume was found in samples supplemented with 20 and 30% non-fermented chia seeds (on average, 39.3 and 41.0% lower, respectively, in comparison with control breads). Samples prepared with SMF and SSF chia seeds showed, on average, 1.74 cm^3^/g specific volume. The highest shape coefficient was attained in breads supplemented with 10% non-fermented chia seeds (2.05). Shape coefficients of other samples were, on average, lower, by 16.1% (control breads and breads prepared with 10% SSF chia seeds), 22.0% (breads prepared with 20 and 30% SMF chia seeds), 36.1% (breads prepared with 20% non-fermented chia seeds), 41.0% (breads prepared with 30% non-fermented chia seeds and with 20 and 30% SSF chia seeds), and 47.3% (breads prepared with 10% SMF chia seeds). In all cases, supplementation with chia seeds resulted in a lower porosity of the bread. However, a correlation between bread specific volume and porosity was not found, and the lowest porosity was obtained in samples with 30% non-fermented chia seeds (61.3%). Porosity showed moderately positive correlations with bread moisture content (r = 0.765, *p* ≤ 0.001) and mass loss after baking (r = 0.701, *p* ≤ 0.001). The latter characteristic was lower in breads prepared with SSF chia seeds in comparison with control samples (on average, 21.0% lower mass loss after baking).

Adamczyk et al. [24] reported that the addition of between 1 and 5% whole chia seeds to wheat bread increased the specific volume. Nevertheless, a decrease in specific volume of breads with ground chia seeds (2.5–7.5%) was observed by Kowalski et al. [29]. The reduction in loaf specific volume of wheat and rice breads with legume mixtures or chia flours was also noted at concentrations greater than 15% [18]. Bustos et al. [19] observed that the inclusion of chia flour (20%) or sourdough (40%) led to a marginal reduction in the specific volume of loaves when compared to wheat bread. According to data gathered by previous researchers, the addition of chia seeds or flour usually results in a lower specific volume of wheat bread. This can be explained by the fact that chia seeds dilute the gluten of wheat flour and reduce the retention of air bubbles [19,28]. This fact also explains the reduced porosity of breads with chia seeds. Moreover, the presence of chia fibres and the formation of protein-lipid complexes may contribute to a decay in bread volume [24]. In our study, the specific volume of breads with fermented chia seeds was higher compared to breads with unfermented seeds in most cases. The use of sourdough has been shown to both decrease and enhance bread volume due to the drop in pH and activation of amylases and proteinases [19]. The increased moisture content of wheat-chia breads and changes in mass loss after baking are related to chia seeds’ ability to absorb water and prevent its loss during baking thanks to their high fibre and mucilage gel contents [24]. The addition of chia seeds to wheat bread has different effects on baking loss. It was reported that smaller amounts of seeds may not affect this parameter, while the inclusion of higher amounts (6% or 8%) significantly lowered the value of bread baking loss [24].

#### 3.2.2. Hardness

The softest bread texture was reached in breads prepared with 10% SSF chia seeds (0.233 mJ) (Table 4). Samples prepared with 10% SMF chia seeds showed a texture hardness value similar to control samples (on average, 0.352 mJ). However, the hardness showed a tendency to increase when the content of chia seeds (non-fermented or fermented seeds) in the bread formula was increased. Significant correlations between bread texture hardness and other bread parameters (porosity, specific volume, moisture content, and shape coefficient) were not established. Differences in the texture hardness values of breads with chia seeds probably occurred due to the weakening of gluten networks by chia seeds, which are rich in dietary fibres and fat. Similar to our results, Coelho et al. [28] and Kowalski et al. [29] reported an increased firmness of wheat breads with chia seeds and flour, respectively. However, other studies revealed a decrease in the hardness of bread with the addition of between 4 and 8% chia seeds [59]. Unlike our results, Bustos et al. [19] reported a decrease in the firmness of bread with chia flour sourdough. In general, differences between studies may occur due to the different amounts and types (whole seeds, ground seeds, flour, or sourdough) of chia seeds used for bread production as well as different bread preparation methods.

#### 3.2.3. Colour Coordinates

The addition of fermented and 30% non-fermented chia seeds to the wheat bread formula reduced the lightness (L*) and yellowness (b*) of the crust in comparison with the control group. All breads with chia seeds (both types) had lower values of the L* and b* coordinates of the crumb. The crust a* (redness) coordinates of supplemented breads were similar to or lower than those of the control group. The redness values of all samples except B_SSF10_ were higher than those of the control group. The colour of the bread is determined primarily by the ingredients put together. Similar to our results, the reduced lightness (L*) of breads with chia flour or seeds was also reported in other studies [18,24,28]. This effect is caused by chia seeds, which contain phenolic compounds that influence the colour of the bread [18]. Similar tendencies in changes in redness (a*) and blueness (−b*) coordinates were also found by Adamczyk et al. [24].

#### 3.2.4. Acrylamide Content

In comparisons between acrylamide concentrations in bread samples, the lowest acrylamide content was found in samples prepared with 30% SSF chia seeds (11.5 µg/kg) (Table 4). However, in comparison with control samples, samples prepared with 20% SSF chia seeds or 10 and 30% non-fermented chia seeds showed, on average, 34.1% lower acrylamide concentration, while samples prepared with 20% non-fermented chia seeds showed, on average, 17.6% lower acrylamide content. In bread samples prepared with 10 and 30% SMF or 20% SSF chia seeds, acrylamide content was similar to the control breads; however, in samples prepared with 10% SSF chia seeds, the acrylamide content was, on average, 1.98 times higher in comparison with control samples. Acrylamide content in bread samples showed moderate positive correlations with the porosity of bread (r = 0.637, *p* ≤ 0.001), moisture content (r = 0.588, *p* ≤ 0.001), and mass loss after baking (r = 0.756, *p* ≤ 0.001). Additionally, acrylamide concentration showed positive correlations with all the tested bread crumb colour coordinates (L* with r = 0.626, *p* ≤ 0.001, a* with r = 0.479, *p* = 0.007, and b* with r = 0.819, *p* ≤ 0.001). Furthermore, significant positive correlations between acrylamide content in bread samples and bread crust L* and b* coordinates were found (r = 0.805, *p* ≤ 0.001 and r = 0.703, *p* ≤ 0.001, respectively).

Free amino acids (mainly asparagine) and reducing sugars (mainly glucose and fructose) are the precursors for acrylamide formation via the Maillard reaction [60]. The harmful effects of acrylamide include possible damage to the nervous system and male reproductive system, the development of cancer, and the ability to induce mutagenic genes [61]. Temperatures higher than 120 °C, low water-activity, and a high level of fibre accelerate the generation of acrylamide by a considerable level during bread baking [62]. In 2013, the European Commission (EC) established an acrylamide indicative value of 80 µg/kg for bread based on wheat flour [63]. The benchmark level set by EU Regulation 2017/2158 for the presence of acrylamide in wheat-based bread is 50 µg/kg [64]. The values of acrylamide obtained in our study did not exceed those set by the European Union (EU). The variation in concentrations of acrylamide in the breads tested in our experiment may be due to chia seed-induced changes in the baking dough and challenges encountered during the sample weighing procedure. The lower concentrations of acrylamide found in breads with chia seeds compared to the control bread can be explained by a low level of reducing sugars and the presence of phenolic compounds in these seeds [65]. These factors inhibit the Maillard reaction during bread baking. The low pH of fermented substrate can also contribute to reduced concentrations of acrylamide in some breads prepared with fermented chia seeds. Numerous studies have suggested that that the reduction of acrylamide formation is more closely linked to gradual reductions in pH than to the use of reducing sugars and free asparagine by microorganisms [61]. However, it was reported that PUFA in chia seeds can promote acrylamide production [65]. Moreover, due to LAB enzymatic activity, changes in fermented chia seeds are also induced. This probably explains the higher concentration of acrylamide in some breads with fermented (SMF and SSF) chia seeds because, after fermentation with *Lp. plantarum,* an increase in PUFA was noticed. Galluzzo et al. [65] compared the acrylamide content in wheat bread containing chia seeds at different concentrations (2, 5, 7, and 10%). The acrylamide concentrations found in his study were considerably higher when compared to our results. The same author reported that breads with chia seeds contained a higher concentration of acrylamide than the control bread, but these differences were not significant. 

#### 3.2.5. Fatty Acid Profile of Produced Bread

The fatty acid profile of the bread samples is shown in Table 5. The FA in control bread samples were predominantly linoleic (C18:2), palmitic (C16:0), and 9-octadecenoic (C18:1) acids (contents were 63.8, 17.0, and 14.0% of the total fat content, respectively). However, the main FA in breads supplemented with chia seeds was α-linolenic acid (C18:3 α). This FA content (in chia seed supplemented breads) ranged from, on average, 55.6% of the total fat content (in bread samples prepared with 10 and 20% SSF chia seeds), to 63.8% of the total fat content (in bread samples prepared with 30% non-fermented chia seeds). However, in the FA profile of the breads prepared with 30% non-fermented chia seeds, the presence of eicosanoic acid (C20:0) was determined. In most cases, a higher content of stearic acid (C18:0) was attained in bread supplemented with chia seeds (except for bread samples prepared with 10% SSF chia seeds).

Comparing the FA profiles of all bread samples, control breads showed higher quantities of SFA and MUFA, as well as omega-6 and omega-9. However, breads enriched with chia seeds exhibited higher contents of PUFA and omega-3. Analysing the bread samples in different bread groups (prepared with non-fermented, SMF, and SSF chia seeds), breads prepared with 10% non-fermented chia seeds showed the lowest SFA content. Nevertheless, they showed the highest MUFA, omega-6, and omega-9 contents in comparison with samples prepared with 20 and 30% non-fermented chia seeds. PUFA content in breads prepared with non-fermented chia seeds was similar (on average, 84.4% of the total fat content). Additionally, the highest content of omega-3 was found in bread samples prepared with 30% non-fermented chia seeds (in comparison with samples prepared with 10 and 20% non-fermented chia seeds, 10.9% higher on average). In comparison between bread sample groups prepared with SMF chia seeds, the lowest SFA and the highest PUFA and omega-3 contents were established in breads prepared with 30% SMF chia seeds. However, the highest MUFA, omega-6, and omega-9 contents were revealed in bread samples prepared with 10% SMF chia seeds. Comparing sample groups prepared with SSF chia seeds, significant differences between the content of saturated, polyunsaturated, and omega-9 FA were not found (on average, their contents were 6.83, 84.1, and 9.08% of the total fat content, respectively).

Differences in the FA composition of tested breads are associated with the addition of unfermented and fermented chia seeds. Obtained results regarding the FA profile in tested breads are similar to those reported by Kowalski et al. [29] and Romankiewicz, D., et al. [59]. Wheat flour contains approximately 80% UFA, the majority of which are omega-6 linoleic and omega-9 oleic acids, whereas this flour consists of approximately 20% SFA, with palmitic acid being the most abundant [66,67]. Elevated total cholesterol and low-density lipoprotein (LDL) levels, an unbalanced composition of the gut microbiota, an increased risk of cardiovascular diseases and metabolic syndromes are significantly linked to SFA [68]. In this research study, incorporating chia seeds to wheat bread formula resulted in diminished levels of SFA, particularly palmitic acid. PUFA are well known for their antioxidant and anti-inflammatory properties. Omega-3 PUFA provide numerous health benefits for the control of heart diseases (reducing blood pressure and increasing arterial wall compliance), diabetes, and the development of cancer, depression, various mental illnesses, and chronic diseases caused by increased inflammation [69]. Consumption of omega-6 linoleic acid is essential for proper development and growth and reduces total and LDL cholesterol levels in the blood [70]. However, excessive intake of omega 6 FA and a decline in omega-3 FA correlates with the worldwide growth of chronic illnesses [71]. When opposed to omega-6 PUFA, omega-3 PUFA are thought to be more anti-inflammatory [72]. Our results demonstrated that incorporating unfermented or fermented chia seeds into wheat bread formula decreased the percentage of omega-6 FA while significantly increasing the total PUFA and omega-3 FA, thus contributing to greater omega-3 consumption.

#### 3.2.6. Volatile Compound Profile of Produced Bread

Volatile compounds in the bread samples (% of the total volatile compounds) are shown in Figure 3 and Appendix A. Moreover, the significance of the analysed factors (treatment type (non-fermented, SMF, and SSF) and quantity of chia seeds) and their interactions for volatile compound formation in bread is given in Table 6.

In comparisons between bread sample groups, the main volatile compounds in all bread samples were 3-methyl-1-butanol, ethanol, hexanoic acid, and phenylethyl alcohol. Analysed factors and their interactions were not significant for the content of the above-mentioned volatile compounds in bread (Table 6). However, caryophyllene, 2-ethyl-3,5-dimethylpyrazine, and benzeneacetaldehyde were only found in bread samples supplemented with chia seeds. Analysed factors and their interactions were significant for caryophyllene and 2-ethyl-3,5-dimethylpyrazine contents in bread samples (*p* < 0.001); however, the type of chia seed treatment was not significant for benzeneacetaldehyde content in bread.

In all bread samples, 2-ethylpyrazine, 1-hexanol, tetradecane, acetic acid, 2-propyl-1-pentanol, benzaldehyde, 2-methylpropanoic acid, 2-methyl-butanoic acid, heptanoic acid, octanoic acid, and 4-vinyl-guaiacol were detected. Most of these compounds were influenced by one or both analysed factors as well as their interactions, except 2-propyl-1-pentanol. Additionally, in control breads and samples prepared with 20% non-fermented or 30% SSF chia seeds, 2-methylpyrazine and 1-(2-furanyl)-ethanone were not detected. These two volatile compound contents were significantly influenced by the quantity of chia seeds used for breadmaking as well as interactions between analysed factors.

3-furaldehyde and 3-furanmethanol were not found in bread samples prepared with 20% non-fermented chia seeds. Analysed factors and their interactions were significant for 3-furaldehyde content in bread. However, the type of chia seed treatment had no significant effect on 3-furanmethanol content in bread. 5-methyl-2-furancarboxaldehyde was not established in control breads or samples prepared with 10 and 20% non-fermented chia seeds or 30% SSF chia seeds. The type of chia seed treatment and interaction between analysed factors showed significant influence on 5-methyl-2-furancarboxaldehyde content in bread. Maltol was found in most of the samples except for the control and breads prepared with 20% non-fermented chia seeds. Analysed factors and their interactions were significant for maltol content in bread samples.

Wheat bread crust, crumb, or both have been found to contain an extensive variety of volatile compounds, such as alcohols, ethers, aldehydes, ketones, esters, pyrazines, acids, furans, hydrocarbons, sulphur compounds, pyrrolines, and lactones [73]. Volatile compounds in bread are produced by non-enzymatic Maillard reactions and caramelization of sugars under increased baking temperatures, as well as enzymatic processes during dough preparation—more specifically, when yeasts and LAB ferment the dough carbohydrates [74,75]. In addition, fermentation with LAB provides different organoleptic notes and enhances the aroma of bread due to peptide hydrolysis, polysaccharide synthesis, and antimicrobial properties [76]. Aldehydes, alcohols, and ketones can result from lipid oxidation, and certain aldehydes can be produced inside yeast cells by the Ehrlich pathway during the breakdown of the amino acids in flour(s) [74]. It was reported that 3-methyl-1-butanol is the most abundant compound in bread, which originates from fermentation (the degradation of L-leucine by yeasts) and is characterised as having balsamic, malty, and alcoholic odours [77]. In our study, this compound also dominated the volatile compound profile of wheat bread, but its concentration decreased with the addition of unfermented or fermented chia seeds.

Ethanol is synthesised during fermentation processes, and it was the second most common volatile compound in the control bread, although its concentration significantly decreased in breads with unfermented and fermented chia seeds. This probably occurred due to the reduced fermentable sugar content in breads with unfermented seeds and the presence of LAB in fermented seeds, which prevent yeast from multiplying in the same manner and hence produce less ethanol [74]. Note that *Lp. plantarum* is a facultative heterofermenter.

Hexanoic acid is produced by non-enzymatic lipid oxidation reactions or fermentation [78]. The fermentation products of the Ehrlich route include phenylethyl alcohol, which is characterised as having a wilted rose and rose-honey-like odour [79]. The lower concentration of this compound in breads with chia seeds may be due to changes in yeast activity. 1-hexanol originates from lipid oxidation or fermentation and is one of the main components of wheat flour; it possesses green grass, woody, flowery, sweet, and mild odours [77]. The higher concentration of this VC observed in most breads with fermented chia seeds can be explained by the increased PUFA content in fermented seeds.

Acetic acid is synthesised during fermentation with facultative or obligate heterofermentative *Lactobacillus* or in yeast cells via the FA synthase pathway [77]. Heterocyclic volatile compounds such as 2-methylpyrazine, 2-ethylpyrazine, and 2-ethyl-3,5-dimethylpyrazine are produced via Maillard and Strecker aldehyde reactions [80], and have roasted, burnt, sweet, popcorn, nutty, and earthy odours. 2-methylpyrazine and maltol are common compounds in wheat bread, while 1-(2-furanyl)-ethanone and 5-methyl-2-furancarboxaldehyde are produced in wheat bread with sourdough [73,81,82,83]. The addition of chia seeds to bread could increase the formation of benzeneacetaldehyde, which has a honey-like odour and results from phenylalanine amino acid degradation [84]. The presence of caryophyllene can be related to the fact that this compound is found in the essential oil of chia leaves [5]. Maltol is characterised as having a caramel-like odour and is recognised as a Maillard flavour compound [85]. Certain lipid oxidation products can be transformed by LAB into corresponding alcohols, and this is the case for 3-methylbutanoic acid, which is produced when aldehyde dehydrogenase oxidises 3-methybutanal [86].

#### 3.2.7. Sensory Properties and Overall Acceptability of Produced Bread

Sensory properties and overall acceptability (OA) of the wheat bread are shown in Figure 4a (colour and odour characteristics), Figure 4b (flavour characteristics), Figure 4c (texture characteristics), and Figure 4d (overall acceptability). The most intense bread colours were attained in samples prepared with 10 and 20% non-fermented chia seeds, 30% SMF chia seeds, and 20 and 30% SSF chia seeds (on average, scores of 6.3) (Figure 4a). The most intense odours were reached in samples prepared with 10 and 20% SMF chia seeds and with 30% SSF chia seeds (on average, scores of 9.0) (Figure 4a). Furthermore, samples prepared with 10 and 20% SMF chia seeds showed the most intense bread odours (on average, scores of 7.1). The most intense additive odours were detected in samples with 30% SMF or SSF chia seeds (on average, scores of 9.7). The most intense flavours were obtained in control samples and samples prepared with 20 and 30% SMF chia seeds (on average, scores of 7.6) (Figure 4b). Control samples possessed the most intense bread flavour, and the most intense additive flavours were observed for bread samples prepared with 20 and 30% SMF chia seeds and with 30% SSF chia seeds (on average, scores of 8.3). The most intense acidity was felt in breads with 30% SSF chia seeds. Analysing bread sensory texture characteristics, the lowest bitterness scores were given for control breads (Figure 4c). The highest porosities were found in control breads and samples prepared with 20% SMF chia seeds or 30% SSF chia seeds (on average, scores of 8.3). The highest brittleness values were found in breads with 10 and 20% SMF chia seeds or with 30% SSF chia seeds (on average, scores of 6.6). Control samples and breads prepared with 10 and 20% non-fermented or SMF chia seeds or 10% SSF chia seeds showed the highest springiness (on average, scores of 8.3). The hardest textures were obtained in samples with 20% non-fermented chia seeds and 20% SSF chia seeds (on average, 8.7 scores). The lowest moisture values were perceived in testing breads prepared with 10 and 20% non-fermented chia seeds or 20% SMF chia seeds (on average, 3.0 scores). Finally, the most acceptable samples for the panel were control breads and breads prepared with 10 and 20% SMF chia seeds (on average, scores of 8.3) (Figure 4d).

It was found that there are moderately positive correlations between bread colour, bread odour, flavour intensity, bread flavour, and the acrylamide concentration in bread (r = 0.471, *p* = 0.009; r = 0.431, *p* = 0.017; r = 0.623, *p* ≤ 0.001; r = 0.405, *p* = 0.026, respectively).

The decrease in sensory attributes and lower scores of acceptability of breads with chia seeds were reported in other studies [28,29,30]. However, Zhu et al. [87] reported that the incorporation of as much as thirty percent chia seeds has little impact on the sensory acceptability and no effect on the overall acceptability of Chinese steamed bread. Sayed-Ahmad et al. [88] found that the sensory attributes of whole wheat-based bread enriched with chia flour were similar to those of the control bread. Compared to the control bread, lower acceptability and significant variations in colour, odour, texture, taste, and appearance of pan bread with 3–12% chia seed powder were observed by Boriy et al. [89]. Such components as polyphenols in chia seeds have a bitter taste, which explains the increased bitterness of tested breads with chia seeds [80]. Breads prepared using LAB were rated better for certain sensory attributes (texture, appearance, and flavour) and acceptability than breads prepared only with yeast [90]. It was reported that the addition of fermented legumes or quinoa to wheat bread maintained good acceptability and improved certain sensory properties of the bread [91,92]. Similar tendencies and even higher overall acceptability were noticed when chia seed flour and flaxseed sourdoughs were used for gluten-free bread production [18].

## 4. Conclusions

Fermentation with *Lactiplantibacillus plantarum* strain No. 122 led to noticeable changes in the characteristics of chia seeds, such as reduced content of biogenic amines and saturated fatty, ω-6, and ω-9 acids, as well as increased levels of ω-3 α-linolenic acid. Addition of non-fermented and fermented chia seeds (10, 20, and 30%) to wheat bread elicited both positive and negative changes in bread quality parameters. The values of acrylamide in bread obtained in this study did not exceed those set by EU regulations. Most of the breads with chia seeds received good overall acceptability scores. Incorporation of non-fermented or fermented chia seeds into wheat bread formulas could contribute to greater ω-3 consumption. Fermentation with *Lp. plantarum* can be recommended in order to improve chia seeds’ nutritional value. Moreover, supplementation of bread with non-fermented or fermented chia seeds at certain levels enhances the fatty acid profile and certain sensory properties and diminishes the acrylamide concentration in wheat bread. However, further research is needed to develop a more appropriate formula for increasing the acceptability of wheat breads with chia seeds while maintaining improved nutritional quality.

## Figures and Tables

**Figure 1 foods-12-02093-f001:**
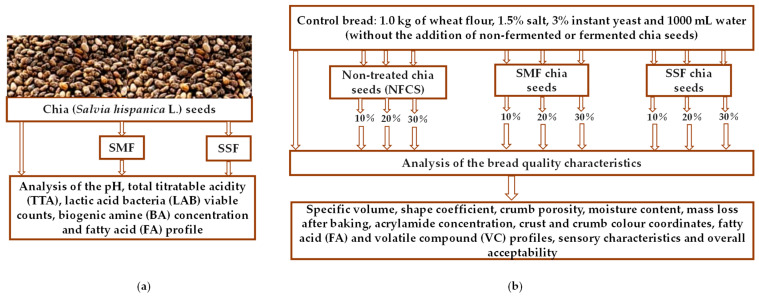
Principal scheme of the experiment: (**a**) pre-treatment and characterisation of chia seeds, (**b**) bread preparation and analysis (NFCS—non-fermented chia seeds; SMF—submerged fermentation; SSF—solid-state fermentation).

**Figure 2 foods-12-02093-f002:**
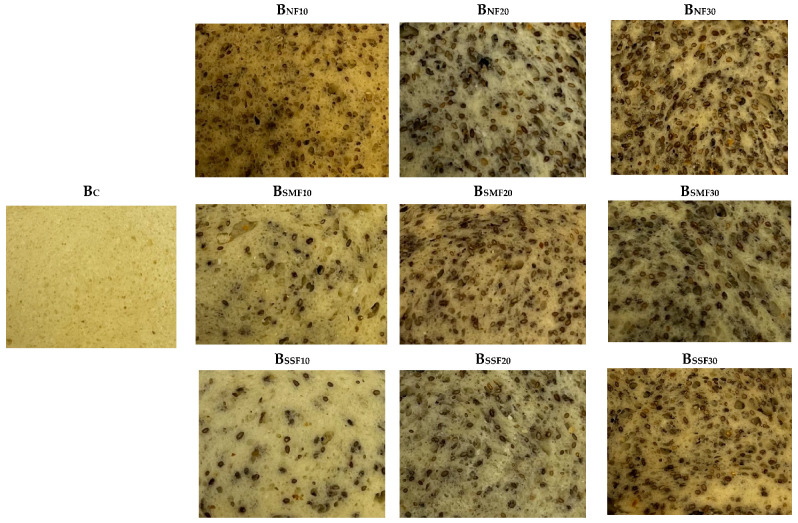
Bread images (B—bread; C—control bread, without chia seeds; 10, 20, 30—amount of the chia seeds added (% of the flour content); NF—non-fermented; SMF—submerged fermentation; SSF—solid-state fermentation).

**Figure 3 foods-12-02093-f003:**
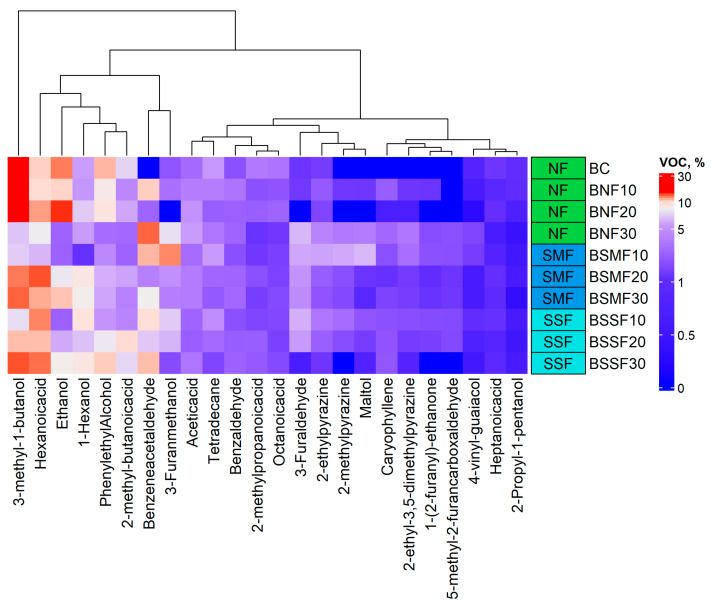
Volatile compounds of the breads with non-fermented and fermented chia seeds (% from the total volatile compounds) (B—bread; C—control bread, without chia seeds; 10, 20, 30—amount of the chia seeds added (% of the flour content); NF—non-fermented; SMF—submerged fermentation; SSF—solid-state fermentation).

**Figure 4 foods-12-02093-f004:**
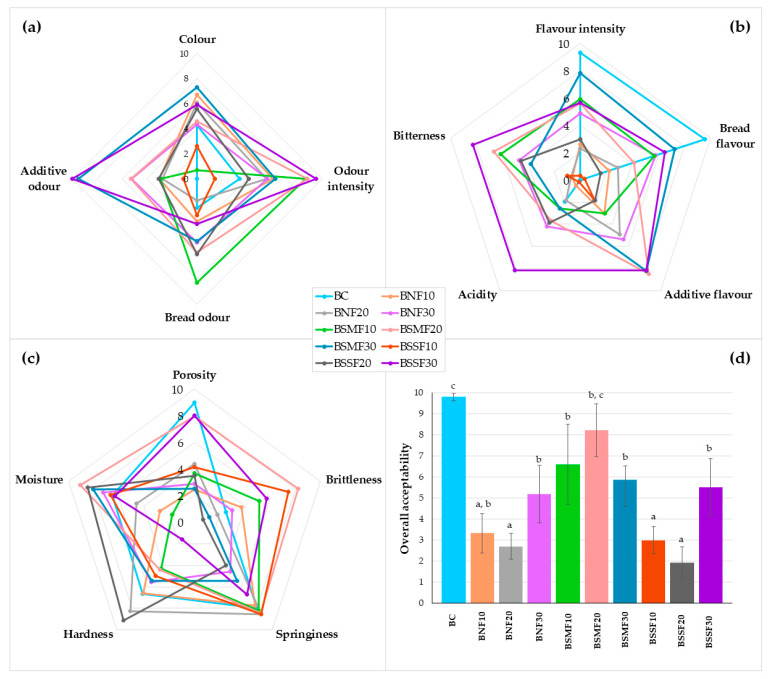
Sensory profile of produced bread: (**a**) colour and odour characteristics, (**b**) flavour characteristics, (**c**) texture characteristics, (**d**) overall acceptability (OA). B—bread; C—control bread, without chia seeds; 10, 20, 30—amount of the chia seeds added (% of the flour content); NF—non-fermented; SMF—submerged fermentation; SSF—solid-state fermentation. a–c Means with different letters are significantly different (*p* ≤ 0.05).

**Table 1 foods-12-02093-t001:** Acidity parameters (pH and TTA) and lactic acid bacteria (LAB) viable counts of chia seeds.

Chia Seed Samples	pH		TTA, °N	LAB Viable Counts, log_10_ CFU/g
	0 h	After 24 h		
C-CS	6.44 ± 0.02	na	na	3.33 ± 0.29 a
SMF-CS		3.75 ± 0.03 a	9.09 ± 0.05 a	8.95 ± 0.31 b
SSF-CS		4.05 ± 0.01 b	11.1 ± 0.08 b	8.60 ± 0.42 b

CS—chia seeds; C—control, non-fermented chia seeds; SMF—submerged-fermented; SSF—solid-state-fermented; TTA—total titratable acidity; °N—Neiman degree; LAB—lactic acid bacteria; CFU—colony-forming units. The data is expressed as mean values (*n* = 3) ± SE; SE—standard error; na—not analysed. a,b Mean values between samples within the lines with different letters are significantly different (*p* ≤ 0.05).

**Table 2 foods-12-02093-t002:** Biogenic amine concentrations (mg/kg) in non-fermented and fermented chia seeds.

Biogenic Amine	Samples		
	C-CS	SMF-CS	SSF-CS
Tryptamine	106.8 ± 6.41 a	107.6 ± 8.23 a	101.8 ± 6.54 a
Phenylethylamine	35.4 ± 3.12	nd	nd
Putrescine	nd	nd	nd
Cadaverine	nd	nd	nd
Histamine	nd	nd	nd
Tyramine	nd	nd	nd
Spermidine	124.3 ± 8.25 b	125.5 ± 9.32 b	96.8 ± 8.45 a
Spermine	31.02 ± 2.71	nd	nd

CS—chia seeds; C—control, non-fermented chia seeds; SMF—submerged-fermented; SSF—solid-state-fermented; nd—not detected. The data is expressed as mean values (*n* = 3) ± SE; SE—standard error; a,b Mean values between samples within the columns with different letters are significantly different (*p* ≤ 0.05).

**Table 3 foods-12-02093-t003:** Fatty acid composition (% of the total fat content) of the chia seed samples.

Fatty Acid	CS-CS	SMF-CS	SSF-CS
	%, of the Total Fat Content		
C16:0	6.91 ± 0.25 a	6.44 ± 0.23 a	6.51 ± 0.47 a
C18:0	3.57 ± 0.02 b	2.60 ± 0.02 a	2.57 ± 0.01 a
C18:1	6.82 ± 0.04 b	4.95 ± 0.04 a	4.94 ± 0.03 a
C18:2	20.6 ± 0.20 c	19.4 ± 0.10 a	19.8 ± 0.02 b
C18:3 α	61.9 ± 0.17 a	66.4 ± 0.23 b	66.0 ± 0.21 b
C20:0	0.24 ± 0.02 b	0.17 ± 0.01 a	0.21 ± 0.02 b
C20:1	0.13 ± 0.01 a	nd	nd
SFA	10.6 ± 0.11 b	9.22 ± 0.09 a	9.29 ± 0.14 a
MUFA	6.94 ± 0.05 b	4.95 ± 0.04 a	4.94 ± 0.25 a
PUFA	82.5 ± 0.35 a	85.8 ± 0.13 b	85.8 ± 0.32 b
Omega-3	61.9 ± 0.14 a	66.4 ± 0.23 b	66.0 ± 0.32 b
Omega-6	20.6 ± 0.16 c	19.4 ± 0.10 a	19.8 ± 0.15 b
Omega-9	6.94 ± 0.07 b	4.95 ± 0.04 a	4.94 ± 0.10 a

CS—chia seeds; C—control, non-fermented chia seeds; SMF—submerged-fermented; SSF—solid-state-fermented; C16:0—palmitic acid; C18:0—stearic acid; C18:1—octadecenoic acid; C18:2—linoleic acid; C18:3 α—α-linolenic acid; C20:0—eicosanoic acid; C20:1—*cis*-11-eicosenoic acid; SFA—saturated fatty acids; MUFA—monounsaturated fatty acids; PUFA—polyunsaturated fatty acids. The data is expressed as mean values (*n* = 3) ± SE; SE—standard error; nd—not detected. a–c Mean values between samples within the columns with different letters are significantly different (*p* ≤ 0.05).

**Table 4 foods-12-02093-t004:** Bread quality parameters: specific volume, shape coefficient, porosity, moisture content, mass loss after baking, acrylamide concentration, and crust and crumb colour characteristics.

Bread Samples	Specific Volume, cm^3^/g	Shape Coefficient	Porosity, %	Moisture Content, %	Mass Loss after Baking, %		Texture Hardness, mJ	
B_C_	2.34 ± 0.06 ^d^	1.69 ± 0.03 ^e^	73.1 ± 0.11 ^h^	39.7 ± 0.06 ^b^	12.6 ± 0.83 ^b^		0.333 ± 0.018 ^b^	
B_NF10_	1.63 ± 0.11 ^b^	2.05 ± 0.05 ^f^	69.4 ± 0.10 ^f^	38.2 ± 0.07 ^a^	15.1 ± 1.21 ^c^		0.633 ± 0.025 ^d^	
B_NF20_	1.42 ± 0.07 ^a^	1.31 ± 0.03 ^c^	65.2 ± 0.06 ^b^	39.7 ± 0.19 ^b^	12.2 ± 0.24 ^b^		1.17 ± 0.03 ^f^	
B_NF30_	1.38 ± 0.13 ^a^	1.18 ± 0.04 ^b^	61.3 ± 0.03 ^a^	41.2 ± 0.10 ^d^	11.1 ± 1.02 ^a,b^		1.37 ± 0.06 ^g^	
B_SMF10_	1.66 ± 0.08 ^b,c^	1.08 ± 0.03 ^a^	69.8 ± 0.05 ^g^	40.7 ± 0.37 ^c^	13.4 ± 1.08 ^b,c^		0.370 ± 0.020 ^b^	
B_SMF20_	1.87 ± 0.08 ^c^	1.57 ± 0.04 ^d^	68.3 ± 0.03 ^d^	41.9 ± 0.39 ^e^	12.5 ± 1.21 ^b^		0.470 ± 0.030 ^c^	
B_SMF30_	1.64 ± 0.13 ^b,c^	1.62 ± 0.02 ^d^	64.6 ± 0.05 ^b^	42.8 ± 0.42 ^f^	10.4 ± 1.05 ^a,b^		1.00 ± 0.10 ^e^	
B_SSF10_	1.75 ± 0.17 ^b,c^	1.74 ± 0.03 ^e^	68.7 ± 0.04 ^e^	42.3 ± 0.31 ^e,f^	9.58 ± 0.91 ^a^		0.233 ± 0.020 ^a^	
B_SSF20_	1.85 ± 0.09 ^c^	1.21 ± 0.02 ^b^	66.1 ± 0.03 ^c^	43.3 ± 0.44 ^f^	10.3 ± 0.73 ^a^		0.667 ± 0.031 ^d^	
B_SSF30_	1.66 ± 0.14 ^b,c^	1.23 ± 0.02 ^b^	64.7 ± 1.14 ^b^	44.1 ± 0.36 ^g^	10.0 ± 0.89 ^a^		0.867 ± 0.042 ^e^	
	Crust			Crumb				Acrylamide concentration, µg/kg
	L*	a*	b*	L*	a*	b*		
B_C_	52.5 ± 1.65 ^c^	11.4 ± 0.43 ^b^	20.7 ± 0.72 ^d^	80.3 ± 0.53 ^i^	−0.393 ± 0.021 ^b^	21.9 ± 0.62 ^h^		26.1 ± 1.77 ^d,e^
B_NF10_	51.8 ± 0.48 ^c^	9.98 ± 0.49 ^a,b^	19.4 ± 0.76 ^c,d^	64.8 ± 0.42 ^e^	0.043 ± 0.003 ^d^	16.7 ± 0.27 ^f^		16.5 ± 1.48 ^b^
B_NF20_	52.8 ± 1.62 ^c^	10.1 ± 0.47 ^a,b^	20.8 ± 0.63 ^d^	60.9 ± 0.54 ^c^	0.637 ± 0.023 ^g^	13.6 ± 0.12 ^d^		21.5 ± 1.05 ^c^
B_NF30_	49.3 ± 1.10 ^b^	9.47 ± 0.35 ^a^	18.2 ± 0.17 ^c^	56.3 ± 0.44 ^a^	1.09 ± 0.07 ^i^	11.9 ± 0.11 ^a^		17.4 ± 0.65 ^b^
B_SMF10_	48.6 ± 1.33 ^b^	11.9 ± 0.43 ^b^	18.9 ± 0.99 ^c^	72.5 ± 0.65 ^h^	−0.117 ± 0.009 ^c^	17.2 ± 0.15 ^g^		28.6 ± 1.19 ^e^
B_SMF20_	43.7 ± 2.15 ^a^	11.3 ± 0.19 ^b^	16.1 ± 0.98 ^a,b^	65.5 ± 0.07 ^f^	0.077 ± 0.006 ^e^	14.7 ± 0.16 ^e^		17.7 ± 1.09 ^b^
B_SMF30_	47.9 ± 1.61 ^b^	8.99 ± 0.65 ^a^	16.4 ± 0.58 ^b^	58.2 ± 0.32 ^b^	0.677 ± 0.012 ^h^	12.5 ± 0.09 ^b^		28.4 ± 0.97 ^e^
B_SSF10_	49.3 ± 0.93 ^b^	11.9 ± 0.36 ^b^	19.5 ± 0.33 ^c^	70.0 ± 0.40 ^g^	−0.567 ± 0.013 ^a^	17.4 ± 0.16 ^g^		51.6 ± 3.50 ^f^
B_SSF20_	49.3 ± 1.18 ^b^	11 ± 0.83 ^b^	18.6 ± 0.30 ^c^	63.5 ± 0.61 ^d^	0.477 ± 0.028 ^f^	14.6 ± 0.13 ^e^		24.5 ± 1.10 ^d^
B_SSF30_	44.1 ± 1.03 ^a^	9.42 ± 0.42 ^a^	15.4 ± 0.11 ^a^	60.2 ± 0.59 ^c^	0.430 ± 0.022 ^f^	12.8 ± 0.15 ^c^		11.5 ± 1.12 ^a^

B—bread; C—control bread, without chia seeds; 10, 20, 30—amount of the chia seeds added (% of the flour content); NF—non-fermented; SMF—submerged fermentation; SSF—solid-state fermentation. L* lightness; a* redness or −a* greenness; b* yellowness or −b* blueness; NBS—National Bureau of Standards units. The data is expressed as mean values (*n* = 3) ± SE; SE—standard error; ^a–i^ Mean values within the lines with different letters are significantly different (*p* ≤ 0.05).

**Table 5 foods-12-02093-t005:** Fatty acid content (% of the total fat content) of the bread samples.

Bread Samples	Fatty Acid Profile (%, of the Total Fat Content)					
	C16:0	C18:0	C18:1	C18:2	C18:3 α	C20:0
B_C_	17.0 ± 0.05 ^f^	0.762 ± 0.021 ^b^	14.0 ± 0.10 ^h^	63.8 ± 0.52 ^g^	4.43 ± 0.11 ^a^	nd
B_NF10_	6.63 ± 0.11 ^c^	1.23 ± 0.09 ^c^	7.73 ± 0.05 ^e^	27.9 ± 0.14 ^e^	56.5 ± 0.36 ^c^	nd
B_NF20_	7.56 ± 0.09 ^e^	1.80 ± 0.07 ^d^	7.07 ± 0.08 ^d^	26.4 ± 0.17 ^d^	57.2 ± 0.41 ^c^	nd
B_NF30_	6.70 ± 0.05 ^c^	2.50 ± 0.09 ^g^	5.32 ± 0.11 ^a^	21.5 ± 0.20 ^a^	63.8 ± 0.58 ^e^	0.160 ± 0.014
B_SMF10_	7.58 ± 0.06 ^e^	1.82 ± 0.07 ^d^	7.09 ± 0.09 ^d^	28.2 ± 0.22 ^e^	55.3 ± 0.43 ^b^	nd
B_SMF20_	7.15 ± 0.05 ^d^	2.04 ± 0.08 ^e^	6.40 ± 0.13 ^c^	25.1 ± 0.23 ^b^	59.3 ± 0.52 ^d^	nd
B_SMF30_	6.62 ± 0.14 ^c^	2.27 ± 0.10 ^f^	5.73 ± 0.18 ^b^	21.5 ± 0.19 ^a^	63.9 ± 0.61 ^e^	nd
B_SSF10_	6.09 ± 0.11 ^b^	0.553 ± 0.019 ^a^	9.12 ± 0.11 ^g^	29.0 ± 0.28 ^f^	55.2 ± 0.54 ^b^	nd
B_SSF20_	5.81 ± 0.12 ^a^	1.12 ± 0.07 ^c^	9.28 ± 0.15 ^g^	27.8 ± 0.25 ^e^	56.0 ± 0.55 ^b,c^	nd
B_SSF30_	5.70 ± 0.16 ^a^	1.19 ± 0.08 ^c^	8.83 ± 0.12 ^f^	25.8 ± 0.23 ^c^	58.5 ± 0.57 ^d^	nd
	Fatty acid profile (%, of the total fat content)					
	SFA	MUFA	PUFA	Omega-3	Omega-6	Omega-9
B_C_	17.8 ± 0.18 ^e^	14.0 ± 0.09 ^h^	68.2 ± 0.59 ^a^	4.43 ± 0.18 ^a^	63.8 ± 0.49 ^g^	14.0 ± 0.08 ^f^
B_NF10_	7.86 ± 0.08 ^b^	7.73 ± 0.08 ^e^	84.4 ± 0.71 ^b,c^	56.5 ± 0.48 ^c^	27.9 ± 0.18 ^e^	7.73 ± 0.11 ^d^
B_NF20_	9.36 ± 0.11 ^d^	7.07 ± 0.09 ^d^	83.6 ± 0.69 ^b^	57.2 ± 0.36 ^c^	26.4 ± 0.14 ^d^	7.07 ± 0.05 ^c^
B_NF30_	9.36 ± 0.14 ^d^	5.32 ± 0.23 ^a^	85.3 ± 0.58 ^c^	63.8 ± 0.41 ^e^	21.5 ± 0.17 ^a^	5.32 ± 0.12 ^a^
B_SMF10_	9.39 ± 0.22 ^d^	7.09 ± 0.08 ^d^	83.5 ± 0.45 ^b^	55.3 ± 0.37 ^b^	28.2 ± 0.21 ^e^	7.09 ± 0.10 ^c^
B_SMF20_	9.19 ± 0.12 ^d^	6.40 ± 0.07 ^c^	84.4 ± 0.32 ^b^	59.3 ± 0.42 ^d^	25.1 ± 0.14 ^b^	6.40 ± 0.25 ^b^
B_SMF30_	8.89 ± 0.09 ^c^	5.73 ± 0.06 ^b^	85.4 ± 0.41 ^c^	63.9 ± 0.51 ^e^	21.5 ± 0.11 ^a^	5.73 ± 0.36 ^a^
B_SSF10_	6.65 ± 0.11 ^a^	9.12 ± 0.11 ^g^	84.2 ± 0.28 ^b^	55.2 ± 0.48 ^b^	29.0 ± 0.16 ^f^	9.12 ± 0.28 ^e^
B_SSF20_	6.92 ± 0.17 ^a^	9.28 ± 0.22 ^g^	83.8 ± 0.47 ^b^	56.0 ± 0.41 ^b,c^	27.8 ± 0.21 ^e^	9.28 ± 0.41 ^e^
B_SSF30_	6.89 ± 0.23 ^a^	8.83 ± 0.17 ^f^	84.3 ± 0.41 ^b^	58.5 ± 0.52 ^d^	25.8 ± 0.09 ^c^	8.83 ± 0.39 ^e^

B—bread; C—control bread, without chia seeds; 10, 20, 30—amount of the chia seeds added (% of the flour content); NF—non-fermented; SMF—submerged fermentation; SSF—solid-state fermentation; C16:0—palmitic acid; C18:0—stearic acid; C18:1—9-octadecenoic acid; C18:2—linoleic acid; C18:3 α—α-linolenic acid; C20:0—eicosanoic acid; SFA—saturated fatty acids; MUFA—monounsaturated fatty acids; PUFA—polyunsaturated fatty acids. The data is expressed as mean values (*n* = 3) ± SE; SE—standard error; nd—not detected. ^a–h^ Mean values between samples within the columns with different letters are significantly different (*p* ≤ 0.05).

**Table 6 foods-12-02093-t006:** Significance of the analysed factors and their interactions for volatile compound formation in bread.

Volatile Compounds	Significance of the Analysed Factors and Their Interactions on Volatile Compounds Formation in Bread		
	Type of Chia Seed Treatment (Non-Fermented, SMF, SSF)	Quantity of the Chia Seeds	Type of Chia Seeds Treatment and Quantity Interaction
Ethanol	0.409	0.108	0.197
3-Methyl-1-butanol	0.250	0.748	0.306
2-Methylpyrazine	0.067	**0.012**	**<0.001**
2-Ethylpyrazine	0.102	**<0.001**	**<0.001**
1-Hexanol	0.077	**0.007**	**0.007**
Tetradecane	0.613	**0.002**	0.431
2-ethyl-3,5-dimethylpyrazine	**<0.001**	**<0.001**	**<0.001**
Acetic acid	**<0.001**	**<0.001**	**0.050**
3-Furaldehyde	**<0.001**	**<0.001**	**<0.001**
2-Propyl-1-pentanol	0.529	0.387	0.429
1-(2-furanyl)-ethanone	0.325	**0.004**	**<0.001**
Benzaldehyde	**0.010**	**0.001**	0.106
2-methylpropanoic acid	**0.016**	**<0.001**	0.123
5-methyl-2-furancarboxaldehyde	**0.003**	0.072	**<0.001**
Caryophyllene	**<0.001**	**<0.001**	**<0.001**
Benzeneacetaldehyde	0.780	**0.018**	**0.018**
3-Furanmethanol	0.169	**0.039**	**0.006**
2-Methyl-butanoic acid	0.051	**<0.001**	0.065
Hexanoic acid	0.810	0.855	0.239
Phenylethyl alcohol	0.263	0.488	0.185
Heptanoic acid	0.657	**0.005**	**<0.001**
Maltol	**0.005**	**0.044**	**0.007**
Octanoic acid	0.977	**0.017**	0.770
4-Vinyl-guaiacol	0.279	**<0.001**	**<0.001**

Influence of the analysed factors and their interaction is significant when *p* ≤ 0.05. Significant values are marked in bold.

## Data Availability

Data are available from the corresponding author upon request.

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
