# Peer review of "Changes in the Physicochemical Properties of Chia (Salvia hispanica L.) Seeds during Solid-State and Submerged Fermentation and Their Influence on Wheat Bread Quality and Sensory Profile"

_foods, 2023, doi:10.3390/foods12112093_

Round 1

Reviewer 1 Report

The authors have performed a research about the effect of two types of fermentation with Lactiplantibacillus plantarum No. 122 strain on the characteristics of chia seeds and their application (10-30 %) to wheat bread production. The research included: determination of acidity, lactic acid bacteria viable counts, biogenic amine, fatty acid profiles of chia seeds; determination of acrylamide, fatty acids, volatile compound profiles, sensory characteristics and overall acceptability of the obtained breads. The manuscript contains valuable results.

The introduction gives a good background of the research described in the manuscript. The experiment is well designed. The methodology is generally well described. Some method descriptions should be added/revised which is mentioned in detailed comments. The obtained results are clearly presented and conclusions are in accordance with the results presented.

Detailed comments can be found below.

Line 164: Please add comment why did you use especially Lp. plantarum No. 122 for the fermentation.

Line 173: Please explain why you added more water to chia seeds for SSF compared to SMF.

Line 186-187: Add comment about expression as Neiman degree.

Line 217: Description of method for determination of bread hardness is missing.

Line 240: Please specify which „data“ you refer to. It is not clearly written. Please explain why you used standard error/standard deviation.

Line 255-257: Split in 2 sentences to increase clarity. Comment about correlation could be in the second sentence.

Line 261-263: Please check this sentence. It seems that pH “drop” in SMF is higher (bigger) compared to SSF sample, but the final pH us lower.  

Line 266: plantarum in lower case

Line 266: Please specify the duration of fermentation.

Line 339: „They“ instead of „he“.

Line 349: To extensive subtitle in my opinion. It would be better to form separate subtitles for all parameters or exclude subtitles.

Line 359-360: It is not clear how the average is calculated as it is stated „most of the samples“. Which samples are included in calculation?

Line 539: Please specify the analysed factors.

Line 544-546: Please explain why you did not include other compounds only present in bread samples with chia seeds, for example 2-methylpyrazine?

Line 684: Conclusion section should be shortened. Authors should focus on overall conclusion rather than repeating the most important results. Also, the authors should add the recommendations and future prospects.

Table 4: Acrylamide concentration and hardness should switch place in table to be in line with the discussion order.

Figure 2: Repetition of same picture Bc is confusing. Please present each picture once. Description for NF abbreviation is missing.

Figure 4: Please uniform sample codes according to the rest of the manuscript. Bitterness is not texture parameter.

S5: In the method description, biscuit sample and cricket flour are mentioned. Please check this.

Please define time after bread baking when samples are analysed.

Please define identification criteria (spectrum matching quality).

Please describe in more detail how and when the retention indexes are used for compound identification.

S6: It is not clear whether the intensity of the sensory attribute or hedonic (liking) score is determined. Based on the defined scale, I conclude you determined hedonic scores for each attribute. However, based on discussion, it seems you determined intensity of each attribute.

Table S1: Please add the description of the term „total volatile compounds“ - it refers to identified compounds only or it includes unknown compounds also?

English should be improved and uniformed (UK/AM style).

Author Response

Reviewer1: The authors have performed a research about the effect of two types of fermentation with Lactiplantibacillus plantarum No. 122 strain on the characteristics of chia seeds and their application (10-30 %) to wheat bread production. The research included: determination of acidity, lactic acid bacteria viable counts, biogenic amine, fatty acid profiles of chia seeds; determination of acrylamide, fatty acids, volatile compound profiles, sensory characteristics and overall acceptability of the obtained breads. The manuscript contains valuable results.

The introduction gives a good background of the research described in the manuscript. The experiment is well designed. The methodology is generally well described. Some method descriptions should be added/revised which is mentioned in detailed comments. The obtained results are clearly presented and conclusions are in accordance with the results presented.

Authors response: Authors are very thankful for the evaluation.

Detailed comments can be found below.

All changes by authors are marked with Word function “tract changes”

Reviewer1: Line 164: Please add comment why did you use especially Lp. plantarum No. 122 for the fermentation.

Authors response: The comment was added: This strain was chosen because Lp. plantarum No. 122 was previously isolated from spontaneous rye sourdough and showed good tolerance to low pH conditions and antibacterial and antifungal activities.

Reviewer1: Line 173: Please explain why you added more water to chia seeds for SSF compared to SMF.

Authors response: Authors are thankful for comment. It was a technical mistake. Corrected: ” …chia seeds / water mass [the ratio 1:5 (w/w), for SSF, the ratio 1:10 (w/w), for SMF].”

Reviewer1: Line 186-187: Add comment about expression as Neiman degree.

Authors response: Comment was added:

The total titratable acidity (TTA) was determined for a 10 g of sample homogenized with 90 mL of distilled water and expressed as the volume, in mL, of 0.1 mol/L NaOH required to achieve a pH of 8.2 (TTA assessed in the Neiman degrees, °N).

Reviewer1: Line 217: Description of method for determination of bread hardness is missing.

Authors response: Description was added in manuscript and Supplementary fileS4 : Bread hardness was determined with Texture Analyser TA.XT2 (StableMicro Systems Ltd., Godalming, UK) as the energy required for the sample deformation. Bread slices of 2 cm thickness were compressed to 10% of their original height at a crosshead speed of 10 mm/s. The resulting peak energy of compression was reported as hardness.

Reviewer1: Line 240: Please specify which „data“ you refer to. It is not clearly written. Please explain why you used standard error/standard deviation.

Authors response: Authors are thankful for comment. There was a technical mistake. All results were expressed as the mean values ± standard error (SE): “The all results were expressed as the mean values (for bread sensory analysis and acceptability n = 30; for the rest of parameters n = 3) ± standard error (SE).”

Reviewer1: Line 255-257: Split in 2 sentences to increase clarity. Comment about correlation could be in the second sentence.

Authors response: Corrected: “Significant differences between LAB count in SMF and SSF samples were not established (on average, LAB count in fermented chia samples was 8.78 log10 CFU/g). A very strong positive correlation between samples TTA and LAB count was found (r=0.946, p=0.004).”

Reviewer1: Line 261-263: Please check this sentence. It seems that pH “drop” in SMF is higher (bigger) compared to SSF sample, but the final pH us lower. 

Authors response: Authors are thankful for the comment. The sentence was corrected: “The lower value of pH and increased LAB viable counts in SMF samples can be explained by the reduced viscosity of the fermentation medium due to a lower solid to liquid ratio when compared to SSF”.

Reviewer1: Line 266: plantarum in lower case

Authors response: corrected.

Reviewer1: Line 266: Please specify the duration of fermentation.

Authors response: The duration of fermentation was 24 hours. This information was included in the sentence.

Reviewer1: Line 339: „They“ instead of „he“.

Authors response: corrected.

Reviewer1: Line 349: To extensive subtitle in my opinion. It would be better to form separate subtitles for all parameters or exclude subtitles.

Authors response: Separate subtitles were formed.

Reviewer1: Line 359-360: It is not clear how the average is calculated as it is stated „most of the samples“. Which samples are included in calculation?

Authors response: This is the average of all samples with SMF and SSF chia seeds. The sentence was corrected into: “Samples, prepared with SMF and SSF chia seeds showed, on average, 1.74 cm3/g specific volume.”

Reviewer1: Line 539: Please specify the analysed factors.

Authors response: Analysed factors were specified: “Moreover, significance of the analysed factors (treatment type (non-fermented, SMF, SSF) and quantity of chia seeds) and their interactions on volatile compound formation in bread is given in Table 6.”

Reviewer1: Line 544-546: Please explain why you did not include other compounds only present in bread samples with chia seeds, for example 2-methylpyrazine?

Authors response: Authors are thankful for a valuable comment and would like to explain that these other compounds were not present in all breads with chia seeds. Please see the following information given in 3.2.6 section:Also, in control breads and samples prepared with 20% of non-fermented and 30% of SSF chia seeds, 2-methylpyrazine and 1-(2-furanyl)-ethanone were not detected. These both volatile compound contents were significantly influenced by quantity of the chia seeds used for breadmaking, as well as analysed factors interaction. Besides, 5-methyl-2-furancarboxaldehyde was not established in control breads and samples prepared with 10 and 20% of non-fermented chia seeds and with 30% of SSF chia seeds. Significant influence on 5-methyl-2-furancarboxaldehyde content in bread was shown by the type of chia seed treatment and factor’s interaction. Maltol was found in most of the samples, except the control and breads prepared with 20% of non-fermented chia seeds. Analysed factors and their interaction were significant on maltol content in bread samples.”

It was reported that these compounds are characteristic volatile compounds of wheat bread (like maltol and 2-methylpyrazine) or wheat bread with sourdough (like 5-methyl-2-furancarboxaldehydel and 1-(2-furanyl)-ethanone).

Reviewer1: Line 684: Conclusion section should be shortened. Authors should focus on overall conclusion rather than repeating the most important results. Also, the authors should add the recommendations and future prospects.

Authors response: Conclusions were improved: “Fermentation with Lactiplantibacillus plantarum No. 122 strain led to noticeable changes in the characteristics of chia seeds, such as reduced content of biogenic amines, saturated fatty, ω-6 and ω-9, and increased levels of ω-3 α-linolenic acid. Addition of non-treated and fermented chia seeds (10, 20 and 30%) to wheat bread elicited both positive and negative changes in bread quality parameters. The values of acrylamide in bread, obtained in this study, did not exceed those set by the EU regulations. Most of breads with chia seeds received good overall acceptability scores. Incorporation of non-treated or fermented chia seeds into wheat bread formula could contribute to greater ω-3 consumption. Fermentation with Lp. plantarum can be recommended in order to improve chia seeds nutritional value. Moreover, supplementation of bread with non-treated or fermented chia seeds at certain levels enhances the fatty acid profile, certain sensory properties and diminishes acrylamide concentration in wheat bread. However, further study is needed to develop a more appropriate formula for increasing the acceptability of wheat breads with chia seeds while maintaining improved nutritional quality.”

Reviewer1: Table 4: Acrylamide concentration and hardness should switch place in table to be in line with the discussion order.

Authors response: Acrylamide concentration and hardness were switched in table.

Reviewer1: Figure 2: Repetition of same picture Bc is confusing. Please present each picture once. Description for NF abbreviation is missing.

Authors response: Description for NF abbreviation was added, and only one picture of Bc was left.

“Figure 2. Bread images (B – bread; C – control bread, without chia seeds; 10, 20, 30 – amount of the chia seeds added (% from the flour content); NF – non-fermented; SMF – submerged fermentation; SSF – solid-state fermentation).”

Reviewer1: Figure 4: Please uniform sample codes according to the rest of the manuscript. Bitterness is not texture parameter.

Authors response: The codes were uniformed and bitterness was added to flavour characteristics.

Reviewer1: S5: In the method description, biscuit sample and cricket flour are mentioned. Please check this.

Authors response: corrected

Reviewer1: Please define time after bread baking when samples are analysed.

Authors response: The bread samples were baked and cooled down for 12 h before the parameters evaluation.

Reviewer1: Please define identification criteria (spectrum matching quality).

Authors response: Minimal spectrum matching criteria was 85 percent.

Reviewer1: Please describe in more detail how and when the retention indexes are used for compound identification.

Authors response: NIST library data contains spectrum and retention index of the analyte. By using retention index and the spectrum matching, more false positive results can be removed.

Reviewer1: S6: It is not clear whether the intensity of the sensory attribute or hedonic (liking) score is determined. Based on the defined scale, I conclude you determined hedonic scores for each attribute. However, based on discussion, it seems you determined intensity of each attribute.

Authors response: The description of sensory analysis was improved: ” Sensory characteristics and overall acceptability of breads were carried out according to the ISO 11136:2014 and ISO 8586:2012 by 30 trained panellists (20 females and 10 males) aged between 20 and 36 years. Panellists were usually consumers of bread. Three 60-min sessions were conducted for the training on the selected terms and the tasting procedure. According to quantitative descriptive sensory analysis, the sensory profile of samples was analysed. The intensity of colour, odour, flavour, acidity, bitterness, porosity, brittleness, springiness, hardness and moisture of the bread were assessed using a 10-point scale, where 0 and 10 indicate the lowest and the highest intensity, respectively. Overall acceptability was evaluated using a 10-point Likert scale ranging from 10 (extremely like) to 0 (extremely dislike). The evaluation was carried out at the sensory laboratory with individual booths and following standard sensory practices. The bread samples were baked and cooled down for 12 h before the sensory evaluation. All samples were coded and served randomly for evaluation. During sensory evaluation, panellists were instructed to drink water or rinse their mouths to clear the palate after each evaluation.”

Reviewer1: Table S1: Please add the description of the term „total volatile compounds“ - it refers to identified compounds only or it includes unknown compounds also?

Authors response: It refers to the identified compounds only. Information was added to the Table S1.

Reviewer1: Comments on the Quality of English Language

English should be improved and uniformed (UK/AM style).

Authors response: English was improved.

Reviewer 2 Report

After reading the manuscript "Changes of chia (Salvia hispanica L.) seed characteristics  during solid-state and submerged fermentation and their in fluence on wheat bread quality", I realized that the manuscript showed in some parts the scientific rigour wanted, but in other parts I have missed it.

The authors have presented critical evaluation only in some paragraphs.

The  objective should be improved.

Thats why I have written some suggestions below in an attempt to improve the paper.

L.1- I wonder if the title and the objective should not mention chemical, physical and sensory analyses 

L.46- Avoid using words that already appear in your title in the keywords as well

L.86- I missed more focus on fibres in the Introduction, only in L.86 it is mentioned, but without much detail or depth. It is difficult to approach "diabetes, colon cancer or chronic cardiovascular diseases" without relating it to chia fibers.

L.132- I believe it is important to be mentioned whether chia can affect the technological and sensory quality of breads as well. "breadmaking process may affect the nutrient stability and bioa vailability as well as formation of other unwanted compounds"

L.135- 146 -The objective needs much improvement. Much of the information that is in these lines can be summarised and the detail will be in Materials and Methods. Some information also looked like part of the abstract.

L.148-  Where did  chia come from?

L.155- We need more details on these formulation ingredients: Wheat flour (whole ? refined ?), salt (Himalayan, refined ?), water (deionised, mineral, room temperature ? warm ?)

L.198- If you would like more information on the type of bread used ( White, Flat, Bioche, Baguette, Ciabatta, Pita ... )  How many batches? Same oven chamber for all  the repetitions?

L.236- Has the project been submitted to an evaluation by a university ethics committee? Did it follow the Helsinki declaration? Please, enter the approval protocol number.

Which sensory test was performed?  How many sessions were conducted during training ? age of the assessors ? were  the assessors usually consumers of this product (bread)  ?  In the supplementary material there is important information that should be in the paper and there are repeated information that should not be there. I suggest a focused review and an evaluation of what is really relevant. For a sensory test like yours, with training, descriptors,  a lot of relevant information was not included, I suggest reading papers and improve this part of your paper. You did not performed accetability  nor traditional attibutes like appearance or color.  I did not understand why. more detais about training sessions are extremely relevant.

L.272- Table 1-  Please, adjust the part written as footnotes with smaller font, this way it looks like table information. In fact, check and improve the footnotes of all the tables, please.

L.306- Table 2 needs to be improved or has it become unconfigured just for me ?

L.684-após melhorar o objetivo, conferir a conclusão.

Minor editing of English language required.

English is always useful to ask a native speaker for a final appreciation.

Author Response

Reviewer2: After reading the manuscript "Changes of chia (Salvia hispanica L.) seed characteristics  during solid-state and submerged fermentation and their in fluence on wheat bread quality", I realized that the manuscript showed in some parts the scientific rigour wanted, but in other parts I have missed it.

The authors have presented critical evaluation only in some paragraphs.

The  objective should be improved.

Thats why I have written some suggestions below in an attempt to improve the paper.

All changes by authors are marked with Word function “tract changes”

Reviewer2:L.1- I wonder if the title and the objective should not mention chemical, physical and sensory analyses 

Authors response: Authors are thankful for a valuable comment. The title and objectives were improved: „Changes in chia (Salvia hispanica L.) seeds physico-chemical properties during solid-state and submerged fermentation and their influence on wheat bread quality”. “The present study aimed to investigate the impact of 24 hours of SSF and SMF with the Lactiplantibacillus plantarum No. 122 strain on the physico-chemical attributes of chia seeds and their addition (10, 20, and 30%) on wheat bread properties.

Reviewer2: L.46- Avoid using words that already appear in your title in the keywords as well

Authors response: Keywords were improved:

“Keywords: Salba-chia; lacto-fermentation; white bread; acrylamide; biogenic amines; fatty acid; volatile compounds”

Reviewer2: L.86- I missed more focus on fibres in the Introduction, only in L.86 it is mentioned, but without much detail or depth. It is difficult to approach "diabetes, colon cancer or chronic cardiovascular diseases" without relating it to chia fibers.

Authors response: Additional information was added: “Chia seeds contain a high amount of insoluble and soluble dietary fibers, whose intake can diminish such health issues as diabetes, coronary heart disease, cancer, and gastrointestinal disorders”.

Reviewer2: L.132- I believe it is important to be mentioned whether chia can affect the technological and sensory quality of breads as well. "breadmaking process may affect the nutrient stability and bioa vailability as well as formation of other unwanted compounds"

Authors response: The sentence was improved: “However, besides improvements in the nutritional profile and antioxidant properties of bread, attention should be paid to the fact that the incorporation of chia seeds and further breadmaking processes may affect the technological and sensory quality of bread, nutrient stability and bioavailability, as well as the formation of other unwanted compounds (e.g., acrylamide, furanic compounds, etc.) [26-30]”

Reviewer2:L.135- 146 -The objective needs much improvement. Much of the information that is in these lines can be summarised and the detail will be in Materials and Methods. Some information also looked like part of the abstract.

Authors response:  Objectives were improved:

“The present study aimed to investigate the impact of 24 hours of SSF and SMF with the Lactiplantibacillus plantarum No. 122 strain on the physico-chemical attributes of chia seeds and their addition (10, 20, and 30%) on wheat bread properties. Fermented chia seeds underwent analysis of acidity, LAB counts, biogenic amine, and fatty acid profile. Produced breads were subjected to assessment of specific volume, shape coefficient, crumb porosity, moisture, mass loss after baking, texture, acrylamide concentration, colour, fatty acid and volatile compound profiles, and sensory properties.”

Reviewer2:L.148-  Where did  chia come from?

Authors response: Chia (Salvia hispanica L.) seeds (composition: protein 21%, fat 31%, total carbohydrates 5%, from which sugars 1%, dietary fibre 34%) were obtained from Urtekram Ltd. (Copenhagen, Denmark).

Reviewer2:L.155- We need more details on these formulation ingredients: Wheat flour (whole ? refined ?), salt (Himalayan, refined ?), water (deionised, mineral, room temperature ? warm ?)

Authors response: The following information was added to 2.3.1. section:  Wheat flour (refined; type 550D; falling number 350 s; wet gluten 27%; ash 0,68%) obtained from Kauno Grudai Ltd. mill (Kaunas, Lithuania), salt (regular, fined table salt ,,O‘Sole“, Szczecin, Poland), and room temperature drinking water (22°C).

Reviewer2: L.198- If you would like more information on the type of bread used ( White, Flat, Bioche, Baguette, Ciabatta, Pita ... )  How many batches? Same oven chamber for all  the repetitions?

Authors response: Type of bread – white bread (wheat bread, prepared from refined wheat flour). Three independent batches were baked. Same oven chamber was used for all the repetitions.

Reviewer2:L.236- Has the project been submitted to an evaluation by a university ethics committee? Did it follow the Helsinki declaration? Please, enter the approval protocol number.

Authors response: the project was not submitted for evaluation by the university's ethics commission, as it is not mandatory, because no animal or human studies were performed. Sensory analysis does not require ethical approval.

Reviewer2: Which sensory test was performed?  How many sessions were conducted during training ? age of the assessors ? were  the assessors usually consumers of this product (bread)  ?  In the supplementary material there is important information that should be in the paper and there are repeated information that should not be there. I suggest a focused review and an evaluation of what is really relevant. For a sensory test like yours, with training, descriptors,  a lot of relevant information was not included, I suggest reading papers and improve this part of your paper. You did not performed accetability  nor traditional attibutes like appearance or color.  I did not understand why. more detais about training sessions are extremely relevant.

Authors response: Authors would like to explain that acceptability and colour assessements were performed. The description of sensory analysis was improved: ” Sensory characteristics and overall acceptability of breads were carried out according to the ISO 11136:2014 and ISO 8586:2012 by 30 trained panellists (20 females and 10 males) aged between 20 and 36 years. Panellists were usually consumers of bread. Three 60-min sessions were conducted for the training on the selected terms and the tasting procedure. According to quantitative descriptive sensory analysis, the sensory profile of samples was analysed. The intensity of colour, odour, flavour, acidity, bitterness, porosity, brittleness, springiness, hardness and moisture of the bread were assessed using a 10-point scale, where 0 and 10 indicate the lowest and the highest intensity, respectively. Overall acceptability was evaluated using a 10-point Likert scale ranging from 10 (extremely like) to 0 (extremely dislike). The evaluation was carried out at the sensory laboratory with individual booths and following standard sensory practices. The bread samples were baked and cooled down for 12 h before the sensory evaluation. All samples were coded and served randomly for evaluation. During sensory evaluation, panellists were instructed to drink water or rinse their mouths to clear the palate after each evaluation.”

Reviewer2: L.272- Table 1- Please, adjust the part written as footnotes with smaller font, this way it looks like table information. In fact, check and improve the footnotes of all the tables, please.

Authors response: Authors are thankful for the comment. Adjusted in all tables (footnotes font 9 pt) according to journal template.

Reviewer2: L.306- Table 2 needs to be improved or has it become unconfigured just for me ?

Authors response: Table 2 was improved.

Reviewer2: L.684-após melhorar o objetivo, conferir a conclusão.

Authors response: Authors cannot answer the question because it is written not in English. However, conclusions were improved.

Reviewer2: Comments on the Quality of English Language

Minor editing of English language required.

English is always useful to ask a native speaker for a final appreciation.

Authors response: English was improved.

Round 2

Reviewer 2 Report

After another evaluation of the manuscript, I realized a great improvement in the quality of the paper, but it still has room for improvement.

The authors have accepted almost all of my sugestions. About the sensory part, you need to insert it in the title and objectives as well, to be  clearer that there is sensory evaluation in your paper. I had already explained it in the first evaluation.

I wondering why you have 6 pages of references and have included important information as supplementary material. Many people end up not reading the supplementary material. Think about it.

Editing of English language is required

Author Response

Reviewer2: After another evaluation of the manuscript, I realized a great improvement in the quality of the paper, but it still has room for improvement.

The authors have accepted almost all of my sugestions. About the sensory part, you need to insert it in the title and objectives as well, to be  clearer that there is sensory evaluation in your paper. I had already explained it in the first evaluation.

Authors response: Sensory part was included in the title and objectives:

Title:Changes in chia (Salvia hispanica L.) seeds physico-chemical properties during solid-state and submerged fermentation and their influence on wheat bread quality and sensory profile”

Objectives: “The present study aimed to investigate the impact of 24 hours of SSF and SMF with the Lactiplantibacillus plantarum No. 122 strain on the physico-chemical attributes of chia seeds and their addition (10, 20, and 30%) on wheat bread properties and sensory profile.”

Reviewer2: I wondering why you have 6 pages of references and have included important information as supplementary material. Many people end up not reading the supplementary material. Think about it.

Authors response: The authors are thankful for comment and would like to explain that information in supplementary files includes method descriptions. The authors believe that readers interested in methods will be able to reach them through supplementary material.

Comments on the Quality of English Language

Editing of English language is required

Authors response: improved.